# Synthesis of Mixed Dinucleotides by Mechanochemistry

**DOI:** 10.3390/molecules27103229

**Published:** 2022-05-18

**Authors:** Faisal Hayat, Mikhail V. Makarov, Luxene Belfleur, Marie E. Migaud

**Affiliations:** Mitchell Cancer Institute, Department of Pharmacology, College of Medicine, University of South Alabama, 1660 Springhill Avenue, Mobile, AL 36604, USA; fhayat@southalabama.edu (F.H.); mimak78@gmail.com (M.V.M.); lbelfleur@southalabama.edu (L.B.)

**Keywords:** riboflavin, thiamine, nicotinamide, nucleotides, dinucleotides, pyrophosphates, mechanochemistry

## Abstract

We report the synthesis of vitamin B1, B2, and B3 derived nucleotides and dinucleotides generated either through mechanochemical or solution phase chemistry. Under the explored conditions, adenosine and thiamine proved to be particularly amenable to milling conditions. Following optimization of the chemistry related to the formation pyrophosphate bonds, mixed dinucleotides of adenine and thiamine (vitamin B1), riboflavin (vitamin B2), nicotinamide riboside and 3-carboxamide 4-pyridone riboside (both vitamin B3 derivatives) were generated in good yields. Furthermore, we report an efficient synthesis of the MW+4 isotopologue of NAD^+^ for which deuterium incorporation is present on either side of the dinucleotidic linkage, poised for isotopic tracing experiments by mass spectrometry. Many of these mixed species are novel and present unexplored possibilities to simultaneously enhance or modulate cofactor transporters and enzymes of independent biosynthetic pathways.

## 1. Introduction

Together, the vitamin B1-thiamine, B2-riboflavin and B3-niacin/nicotinamide derived cofactors include but are not limited to thiamine monophosphate, pyrophosphate and triphosphate, thiamine adenine dinucleotide (ThMP, ThDP, ThTP and ThAD), riboflavin monophosphate and riboflavin adenine dinucleotide (FMN and FAD) and nicotinamide adenine dinucleotide (NAD^+^), respectively. The recent resurgence of basic and biomedical research effort that seeks to probe the mechanisms that these cofactors regulate in cells, tissues and whole organisms [1] has stemmed from new capabilities to manipulate and trace their intracellular abundance; e.g., [2,3]. Methods to boost cofactor levels include the use of specific precursors and the manipulation of the biosynthetic pathways that lead to their conversion to the phosphate-containing cofactors [4,5]. Precursors and cofactors that are modified with stable isotopes are used in metabolomic tracing and flux experiments to inform on the (de)regulation of biosynthetic pathways, and cellular or tissues distribution of these cofactors; e.g., [6,7]. These strategies help better understand why these vitamins and their related cofactors are so central to metabolism and cellular machinery. 

Although much progress has been made on defining the biology of the multiple derivatives of B-vitamins that constitute the B-vitaminome, much remains to investigate regarding how cells transport vitamin B1, B2 and B3 derived cofactors and cofactor precursors across membranes. Specific transporters on the cell surface recognize and import the non-charged water-soluble vitamins that are then rapidly phosphorylated in the cytosol or exported back in the extracellular space by equilibrative transporters [8,9,10,11]. While transporters of extracellular phosphorylated cofactor precursors remain under intense scrutiny, specific mitochondrial membrane carriers, SLC25A19, SLC25A32, and SLC25A51, have been shown to import across the mitochondrial membrane from the cytosol ThDP, NAD and FAD, respectively, and support the sub-cellular distribution of each cofactors [12,13,14] in this organelle. Overcoming vitamin B1, B2 and B3 cofactor depletion in the mitochondrion has been the focus of much basic and clinical research in the past few years [15,16,17,18,19,20,21,22]. Mutations in either of the mitochondrial transporters cause pathologies associated with mitochondrial dysfunction [23]. Yet, rarely has there been reports of simultaneous mutations of these transporters. This raises the possibility for uptake of unnatural payloads to ease mitochondrial dysfunction. This has been used with great success in folate (vitamin B9) biology but rarely considered for cofactors derived from vitamin B1, B2 or B3 [24]. Mixed dinucleotides that could cross membranes by piggy-backing on each other’s transport mechanisms could be valuable to try and overcome pathophysiology associated with mitochondrial transport deficiencies related to phosphate-derived cofactors.

Thiamine is found in a di- and tri-phosphate form in cells, while riboflavin and nicotinamide-derived nucleosides are monophosphorylated and converted to their respective adenine dinucleotide cofactors [4]. Discoveries of the biological functions of adenosine-modified vitamin B1, B2 and B3 derived cofactors keep on expanding as they are enabled by new chemistry e.g., [2,14,15]. However, one major chemical limitation to such advancement is the difficulty in generating the pyrophosphate linkage (P-O-P) that is a core feature of each one of these vitamin-derived cofactors [25]. While several methods to generate P-O-P bonds have been reported in the literature, many fail to report isolated yields of final dinucleotide products or report low conversions and difficulties in isolating products. We reported that mechanochemistry was particularly effective at generating pyrophosphate bonds from adenosine phosphomorpholidate and nucleotides and sugar phosphomonoesters in moderate to good, isolated yields [26]. When we explored more comprehensively the scope of mechanochemistry for vitamin B1, B2 and B3 derived cofactor synthesis, we, and others, observed that not all hydroxyl and phosphate precursors shared the same reactivity and thus were not all amenable to the same synthetic sequences [12,27,28]. Further, under some instances, phosphoimidazolates proved to be more versatile and amenable to unsymmetrical P-O-P bond formation than phosphomorpholidates.

Mechanochemistry is an alternative to solution chemistry and identified in 2019 by IUPAC as one of the 10 world-changing technologies for sustainable chemistry [29]. Mechanochemistry offers the possibility to dramatically reduce the use of bulk solvents and the generation of waste, while still enabling a broad range of organic reactions [30]. It offers molecules with a different reaction environment previously not accessible in solution. Here, we demonstrate that thiamine, a poorly soluble synthetic precursor is easily functionalized under milling conditions since mechanochemistry allows for solvent-free chemistry and favors phosphorylation over chlorination of the primary alcohols. Milling conditions also compared favorably to solution phase conditions for the phosphorylation of ribosylated nicotinamide and are particularly suited for 3-carboxamide-4-pyridone riboside’s phosphorylation, for which no phosphorylated product could be detected in solution phase chemistry. Finally, while exploring the reactivity of riboflavin towards phosphorylating and conjugation reagents, we identified conditions that readily favored the formation of cyclic-FMN, a yet-to-be explored intracellular riboflavin metabolite [31]. Having identified the reactivity patterns of each species, we converted each phosphate to the imidazolated species where possible, conducted mechanochemistry-enabled pyrophosphate bond formation and obtained the symmetrical and asymmetrical dinucleotides in moderate to good, isolated yields. In addition to known species produced in good yields, we were able to generate some novel combinations of vitamin B1, B2 and B3 within the pyrophosphate scaffold (Figure 1). Crucially, as it proceeds in good yields, the mechanochemical approach to dinucleotide bond formation was amenable to the generation of the MW+4 NAD isotopologue via a C5-modified MW+2 adenosine and nicotinamide riboside.

## 2. Results and Discussion

### 2.1. Synthesis of Monophosphates

Several chemical and chemoenzymatic methods are available to access the monophosphoester of nucleosides [32] such as adenosine, and of thiamine, riboflavin, and nicotinamide riboside [33], abbreviated AMP, ThP, FMN and NMN, respectively. However, the water content of the starting materials often compromises the efficacy of the chemical conversions, regardless of the method employed. One approach to overcome this limitation and phosphorylate the C5’ position of canonical nucleosides without prior protection of the C2’, C3’ riboside hydroxyls is the Yoshikawa phosphorylation [34,35]. Trimethyl and triethylphosphate are often used as co-solvents along with a molar excess of POCl_3_ to bring polar nucleosidic reagent in solution, and preferentially activate the C5’ hydroxy position to favor a regioselective phosphorylation. Alternatively and to limit polyphosphorylation of the nucleoside, the use of a mixture of excess POCl_3_, pyridine and water has also been implemented [35]. However, this approach can lead to the partial decomposition of starting materials. Removal of excess phosphotriester solvents is often the plight of synthetic chemists seeking to use the resulting phosphomonoester product in a pyrophosphate bond formation. The inorganic phosphate must also be comprehensively removed, as its presence undermines the subsequent chemistry. Having already established that nicotinamide riboside was particularly amenable to regiospecific phosphorylation at the C5’ position, under mechanochemical conditions that minimized nucleosidic bond breakage upon work up [27], we sought to apply these phosphorylation conditions to thiamine, riboflavin, nicotinamide riboside, 3-carboxamide 4-pyridone riboside and adenosine (Table 1).

The outcomes of the phosphorylation of thiamine (Th), riboflavin (RF), nicotinamide riboside (NR), 3-carboxamide 4-pyridone riboside (PYR) and adenosine (Ade) conducted in solution were compared with the outcomes of phosphorylation conducted under milling conditions (Table 1). Figure 1 summarizes the chemistry applied to the nucleoside scaffolds described in Table 1. Under mechanochemical ball-milling conditions (BM), POCl_3_ (4 eq.) was the sole reagent utilized in the preparation of these monophosphates. The progress of the milling reactions was initially monitored by ^31^P-NMR at 30 min intervals. Milled-reactions were stopped after 60 min while solution reactions were pursued for 24 h if deemed incomplete after the first 60 min. Table 1 reveals that mechanochemical conditions (@30 Hz) were well suited for the conversion of PYR, NR, and thiamine (e.g., Figure 2) to the respective monophosphates in good, isolated yields, but unsuitable for riboflavin, that remained unreactive. ^1^H-NMR analyses of the phosphorylation of adenosine under mechanochemical conditions revealed a mixture of unreacted adenosine (64%) and product AMP (36%), with better yields consistently achieved by solution phase chemistry (Table 1).

### 2.2. Synthesis of Pyrophosphates

We originally reported [26] the two-step synthesis of unsymmetrical and symmetrical dinucleotides, including NAD (**20**) and Ap_2_A (**21**). Since then, several vibrational or mechanical ball milling and solution phase procedures that enable the generation of symmetrical and non-symmetrical dinucleotides have been described and reviewed [36,37,38,39]. This chemistry employed morpholidate precursors generated via solution phase chemistry and purified before use in pyrophosphate bond formations enabled by mechanochemistry. While seeking to generate the morpholidate intermediates of AMP, NMN, PYRMP, ThP and FMN, necessary for the P-O-P bond forming process developed for milling conditions, we rapidly identified several issues that promoted the homodimerization of the phosphate monoesters to the symmetrical pyrophosphates upon activation. It became clear that to access unsymmetrical pyrophosphates in good yields, an alternative procedure had to be developed. More recently, Appy et. al. reported the efficient synthesis of symmetrical dinucleotides in good yields in a two step-one pot approach employing milling and in situ activation of nucleotides by CDI in the presence of imidazole [37]. However, these conditions were not suited to the generation of unsymmetrical dinucleotides. Therefore, we sought conditions that generated the imidazolate of each phosphoester quantitatively, so that these intermediates could then be applied in the syntheses of asymmetrical pyrophosphates in a step-wise one-pot process. Ball-milling conditions proved particularly amenable to this process.

In this approach, individual monophosphates (1 eq.) were ball-milled with CDI (4 eq.) for 1 h at 30 Hz in a stainless-steel jar in the presence of acetonitrile (0.3 μL/mg of monophosphate). ^31^P-NMR spectra of the reaction mixture showed single peaks in the region spanning between −8.5 and −9.7 ppm, indicative of phosphoimidazolate. This conversion was quasi quantitative for all phosphate monoesters, except for FMN that remained unreactive (Table 2). The individual imidazolates still contained in the milling jar were then used in the pyrophosphate bond formation step (Figure 1, step 3). Two methods were then explored for the pyrophosphate bond formation, Method A [25] and Method B [37]. According to Method A, the second monophosphate (1 eq.) unit was immediately added to the jar containing the imidazolate intermediate, along with MgCl_2_.6H_2_O (1.5 eq.), tetrazole (2 eq.), and H_2_O (6 eq.). The jar was then vibrated for 90 min, while according to Method B, the imidazolate containing jar was charged with the second monophosphate in the free acid form with ACN (0.6–0.95 µL/monophosphate) and vibrated for 2 h. The reaction progression was monitored by ^31^P-NMR. Overall, the crude yields obtained for the dinucleotides synthesized using Method B were relatively low, and in most cases, unreacted imidazolate starting materials were recovered. When using Method A, the same dinucleotides were obtained with good yields. The imidazolate intermediates were either converted to the symmetrical or asymmetrical dinucleotides. The crude pyrophosphate reaction mixtures generated using this process were lyophilized, diluted in the least amount of water possible, adsorbed on silica gel, and purified using reversed-phase chromatography (Teledyne) with a C18 column and acetonitrile: pH 7 buffer, 100 mM ammonium acetate (2:98 to 10:90, *v*:*v*). Isolated yields ranged between 12% and 63%.

Overall, combining the in-situ generation of imidazolate by milling with a stepwise activation of the imidazolate intermediate for conjugation with a free monophosphate provided a versatile approach to unsymmetrical pyrophosphates (Table 2). Crucially, the reactivity of the individual imidazolate towards other monophosphate partners and the outcomes of the pyrophosphate forming reaction could be manipulated by choosing to activate one phosphate over the other. Thus, we generated nicotinamide adenine dinucleotide (NAD) (**20**), 3-carboxamide-4-pyridone adenine dinucleotide (4-ox-NAD) (**23**), adenosine and nicotinamide thiamine diphosphate (AThDP and NRThDP (**30a** & **31**), respectively, Figure 3), and nicotinamide riboside riboflavin diphosphate (NMN-FMN) (**33**) (Figure 4). Finally, by using TDP (**36**) instead of AMP (**11**), we also accessed adenine thiamine triphosphate (AThTP) (**37**), in good, isolated yields (Figure 5). Isolated symmetrical byproducts included diadenosine pyrophosphate (AP_2_A) (**21**), dinicotinamide riboside pyrophosphate (NMN-NMN) (**22**), dithiamine pyrophosphate (ThP_2_Th) (**30b**), and di-3-carboxamide-4-pyridone riboside pyrophosphate (di-4-PYR-MP) (**24**). However, even under the conditions that proved to be most successful for the formation of pyrophosphates, FMN (**32**) did not yield riboflavin containing dinucleotides, but instead afforded the cyclic-riboflavin monophosphate, cFMN (**34**) [40,41], as the major product. 

The versatility of this approach proved to be highly valuable when seeking to generate an isotopically labeled NAD in good yields. 

### 2.3. Synthesis of Labeled NAD (**25**)

Recent efforts towards understanding how the vitamin B3 derived cofactors are transported across membranes based on mass spectrometry enabled isotopic tracing metabolomics has required the synthesis of stable isotopically labeled NAD (**25**) [2]. Such species informed on whether NAD was transported intact or was first metabolized to a non-phosphorylated species prior to transport from one organelle to another [2,42]. In the latter case, a NAD isotopologue possessing heavy atoms on either side of the pyrophosphate bonds, for example on each riboside unit, would lead to NAD species possessing lower molecular weight as scrambling would occur, while an NAD transporter would enable the detection of the synthesized NAD isotopologue. While the approach demonstrated that a mitochondrial transporter was indeed responsible for the NAD import in the mitochondrion from the cytoplasm, the synthetic route to the isotopologue was low yielding. Therefore, we sought an approach that could be easily applied to other NAD isotopologues while also useful for a wider range of nucleotides. Crucially, the synthetic commitment to the introduction of isotopes across the building blocks requires the final pyrophosphate formation step to be high yielding, and the synthetic sequence enabled by mechanochemistry offered such prospect. 

Thus, we generated the MW+2 adenosine (**48**) for which deuterium isotopes have been introduced on the C5 position of D-ribose, and the MW+2 nicotinamide riboside isotopologue (**54**) with ^18^O modified nicotinamide. The C5-deuterated adenosine monophosphate was produced from D-ribose (**38**). Following a 2,3-*O*-isopropyledene protection (**39**), followed by oxidation of the C5-primary alcohol to the carboxylic acid and methyl esterification, the ester (**41**) (Figure 6) was isolated in 51% yield following chromatographic purification. Two distinct approaches were explored to introduce deuterium at the C5 position. Reduction of the methyl ester with LiAlD_4_ in diethyl ether occurred in 35% yield, whereas reduction with NaBD_4_ in D_2_O (99.9%) yielded the MW+2 riboside (**42**) in 64% isolated yield. Following the aq. TFA. deprotection of the isopropyledene and removal of excess reagents, the crude oily ^2^H_2_C5-ribose (**43**) was treated with methanol in the presence of conc. H_2_SO_4_. The resulting ^2^H_2_C5-C1-*O*-methyl ribofuranoside (**44**) was converted to the deuterated acetylated ribofuranoside (**46**) with acetic anhydride in glacial acetic acid in the presence of a catalytic amount of concentrated sulfuric acid initially at 0 °C and then room temperature. Longer reaction time increased the formation of pyranoside byproducts. The anomeric mixture of C5 deuterated riboside tetraacetate (**46**) was obtained in 69% isolated yield and did not require any chromatographic purification. Glycosylation of riboside (**46**) with adenine under Vorbrüggen conditions produced the C5-deuterated triacetate adenosine (**47**) in 60% isolated yields. The nucleoside was then treated with a 2:1 combination of MeOH and aq. NH_4_OH for 24 h at 8 °C. After complete deacetylation, the mixture was placed at −20 °C for a week to precipitate the beta isomer of the C5-deuterated adenosine (**48**) in a 25% yield as a white powder. The filtrate was concentrated and stored for additional purification. The phosphorylation of the beta-anomer of C5-deuterated adenosine (**48**) was conducted under solution phase conditions using POCl_3_ in (MeO)_3_PO at −20 °C for 24 h. Upon reaction completion and addition of water, the pH of the solution was raised to 7 by adding triethylamine. The C5-deuterated adenosine monophosphate triethylammonium salt (**49**) (80% yield) was then isolated upon removal of the water by freeze-drying. 

The MW+2 ^18^O labeled NR (**54**) was synthesized according to described procedures [3] and ^18^O-NMN (**15**), prepared by phosphorylation of ^18^O-NR, was activated with CDI under milling conditions. C5-deuterated adenosine monophosphatetriethyl amine salt (**49**) was added to the resulting NMN-derived imidazolate (**19**) and milled to yield a mixture of MW+4 labeled NAD (**25**) and MW+4 labeled double-NMN (**26**) (Figure 7). According to the ^31^P NMR spectra of the crude composition, mixture of NAD (**25**) and double NMN (**26**) represented 29%, with 71% of remaining labeled AMP and NMN that could be recycled. As such, the crude material was diluted in a small amount of water, adsorbed in silica, and purified by reverse phase chromatography to yield pure M+4 NAD (**25**) and pure M+4 double NMN (**26**) in 8% and 11% isolated yields, respectively, as quantified by UV spectrometry, while the mixture of unreacted labeled AMP (**14**) and NMN (**15**) was recovered. 

We presented in detail how a two-step one pot approach could be applied to the generation of unsymmetrical canonical and non-canonical dinucleotides, obtained in isolated, UV-quantified yields that compare well with the yields of the current protocols of simpler dinucleotides. With milling, we gathered a greater understanding of the phosphorylation step of the primary alcohol of each synthetic components and altered our approach to the pyrophosphate bond formation to minimize homodimerization. Crucially, we observed that thiamine, a vitamin particularly notorious for its poor solubility in most solvents, was particularly amenable to mechanochemical conditions, both for phosphorylation and for further conversions. Alternatively, we observed that the mechanochemical conditions we explored were not well suited for the synthesis of FMN (**32**) and FAD (**35**) from FMN (**32**), as riboflavin proved particularly unreactive towards POCl_3_, and divalent cations promoted cyclic-FMN (**34**) [41] formation in very good yields, instead. Finally, while the challenges of achieving good isolated yields following chromatography purifications still remain, we observed that conditions where nucleosides and nucleotides could be exhaustively dried offered better outcomes. We had made similar observations when handling P(III) reagents with nucleosides [43,44]. Yet, when we applied this synthetic approach to isotopically labeled NAD nucleoside precursors, we were able to generate an NAD isotopologue in good isolated yields; in quantities that would be well suited to inform on exogenous NAD turn-over, including NAD’s use by sirtuins, PARP enzymes, glycohydrolases, as well as reductases, in cell based-assays by mass spectrometry. 

## 3. Materials and Methods

### 3.1. General Synthetic Procedure for the Preparation of Monophosphates (**11–15** & **28**)

#### 3.1.1. By Mechanochemical Ball-Milling 

A mixture of nucleoside (1 eq.) and POCl_3_ (4 eq.) was introduced into a stainless-steel jar (1.5 mL) along with one stainless steel ball (20 mm in diameter). The reaction vessel, along with another identical empty vessel, was closed and fixed on the vibration arms of a Retsch MM400 miller and vibrated at 30 Hz at room temperature for one hour. After completion of the reaction, the jar was allowed to cool down to room temperature and the contents of the jar were dissolved in a minimum amount of water and stirred at room temperature for 3–4 h for the complete hydrolysis of dichloromonophosphate to dihydroxy monophosphate. The percentage conversion was monitored by ^1^H-NMR (Table 1).

#### 3.1.2. By Solution-Phase Synthesis 

A solution of a specific nucleoside (1 eq.) in trimethyl phosphate (5 mL) was mixed with POCl_3_ (3 eq.) at −5 °C. The mixture was stirred at the same temp for 1 hour and then was kept in the −20 refrigerator for the next 48 h. The reaction progress was monitored by ^1^H and ^31^P-NMR analysis. Upon completion of the reaction, the crude material was treated with MeCN–Et_2_O (1:3), and the white precipitate was separated from the liquor. Once isolated, the solid was stirred with a minimum amount of water overnight for complete hydrolysis of the C5-dichloromonophosphate to the C5 dihydroxymonophosphate [1]. This crude was further purified through the Dowex 50WX8 resin (H+ form) column by using water and formic acid (100%:0% to 90%:10%) as an eluent to give the monophosphate in the isolated yield presented in Table 1. The C5 deuterated adenosine (**14**) was converted into its TEA salt (**49**) and used for the next step without further purification. Among the series, the formation of AMP (**11**), C5 deuterated adenosine (**14**), O^16^ and O^18^ NMN (**12** and **15**) were completed in 48 h, but the conversion of 4PYR to 4PYRMP (**13**) was unsatisfactory. Phosphorylation of thiamine and riboflavin in solution phase were also unsuccessful. 

Adenosine monophosphate (AMP) (**11**): ^1^H-NMR (400 MHz, D_2_O), δ, ppm: 8.53 (s, 1H, Ar-H), 8.33 (s, 1H, Ar-H), 6.01 (d, *J* = 5.2 Hz, 1H, H-1_ribose_), 4.67 (brs, 1H, H-3_ribose_), 4.30 (brs, 1H, H-4_ribose_), 4.06–4.04 (m, 2H, H-5_(A&B)ribose_); ^13^C-NMR (100 MHz, D_2_O), δ, ppm: 149.90, 148.19, 144.91, 142.19, 118.29 (Ar-C), 87.89 (C-1_ribose_), 84.18 (d, *J*_CP_ = 8.6 Hz, C-4_ribose_), 74.67 (C-2_ribose_), 70.22 (C-3_ribose_), 64.25 (d, *J_CP_* = 4.92 Hz, C-5_ribose_); ^31^P-NMR (162 MHz, D_2_O), δ, ppm: 0.18.

Nicotinamide mononucleotide (NMN) (**12**): ^1^H-NMR (400 MHz, D_2_O), δ, ppm: 9.37 (s,1H,Ar-H), 9.19 (d, *J*= 5.2 Hz, 1H, Ar-H), 8.89 (d, *J* = 7.88 Hz, 1H, Ar-H), 8.21 (t, *J* = 6.92 Hz, 1H, Ar-H), 6.13 (d, *J* = 5.16 Hz, 1H, H-1_ribose_), 4.55 (brs, 1H, H-4_ribose_), 4.47 (d, *J* = 4.18 Hz, 1H, H-2_ribose_), 4.35 (brs, 1H, H-3_ribose_), 4.21 & 4.06 (AB part of ABX system, 2H*, J_AB_* = 11.76, *J_AX_* = 3.92, *J_BX_* = 3.92, H_5A-ribose_ & H_5B-ribose_); ^13^C-NMR (100 MHz, D_2_O), δ, ppm: 165.81 (CO), 145.96, 142.46, 139.85, 133.92, 128.51 (Ar-C), 99.94 (C-1_ribose_), 87.35 (d, *J*_CP_ = 8.8 Hz, C-4_ribose_), 77.70 (C-2_ribose_), 70.93 (C-3_ribose_), 64.14 (d, *J*_CP_ = 4.88 Hz, C-5_ribose_); ^31^P-NMR (162 MHz, D_2_O), δ, ppm: −0.11.

3-Carboxamide-4-pyridone riboside monophosphate (4PYRMP) (**13**): ^1^H-NMR (D_2_O), δ, ppm: 8.70 (d, 1H, *J* = 2.1 Hz, H-2_4py_), 8.15 (dd, 1H, *J* = 7.6 Hz, *J* = 2.1 Hz, H-6_4py_), 6.80 (d, 1H, *J* = 7.6 Hz, H-5_4py_), 5.60–5.64 (m, 1H, H-1_ribose_), 4.68 (OH, NH_2_, overlapped with residual water in D_2_O), 4.33–4.34 (m, 1H, H-4_ribose_), 4.27–4.31 (m, 2H, H-3&2 _ribose_), 4.13 (ddd, AB part of ABX system, 1H, *J_AB_* = 11.7 Hz, *J_BX_* = 2.5 Hz, *^3^J_HP_* = 4.4 Hz, H_5A-ribose_), 4.07 (ddd, AB part of ABX system, 1H, *J_AB_* = 11.7 Hz, *J_AX_* = 2.8 Hz, *^3^J_HP_* = 5.4 Hz, H_5B-ribose_), 3.24 (s, 2.5H, MeOH). ^13^C-NMR (D_2_O), δ, ppm: 174.97 (CO_4py_), 166.23 (COco_NH2_), 143.13 (C2_4py_), 140.10 (C6_4py_), 118.28 (C5_4py_), 117.88 (C3_4py_), 97.44 (C-1_ribose_), 85.19 (d, *J*_CP_ = 8.6 Hz, C-4 _ribose_), 76.12 (C-2 _ribose_), 70.25 (C-3 _ribose_), 65.11 (d, *J*_CP_ = 4.8 Hz, C-5 _ribose_). ^31^P-NMR (D_2_O), δ, ppm: –0.18. HRMS, *m*/*z*: found 351.05787 [M + 1], calculated for C_11_H_16_N_2_O_9_P [M + 1] 351.05879.

Thiamine monophosphate (**28**): ^1^H-NMR (400 MHz, D_2_O), δ, ppm: 9.56 (s, 1H, Ar-H), 7.84 (s, 1H, Ar-H), 5.45 (s, 2H, CH_2_-N), 4.02–3.98 (m, 2H, -CH_2__-_CH_2_-P), 3.19 (t, *J* = 5.52 Hz, 2H, -CH_2__-_CH_2_-P), 2.51 (s, 3H, -CH_3_), 2.43 (s, 3H, -CH_3_);^13^C-NMR (100 MHz, D_2_O), δ, ppm: 163.20, 162.95, 154.91, 144.20, 143.10, 135.92, 106.34 (Ar-C), 63.81 (d, *J*_CP_ = 5.06 Hz, -CH_2__-_CH_2_-P), 49.82 (s, 2H, -CH_2-_N), 27.71 (d, *J*_CP_ = 7.52 Hz, -CH_2__-_CH_2_-P), 20.89 (s, 3H,-CH_3_), 11.00 (s, 3H,-CH_3_); ^31^P-NMR (162 MHz, D_2_O), δ, ppm: −0.19.

#### 3.1.3. Synthesis of FMN (**32**)

In a round-bottom flask, POCl_3_ (16 mL) was added to 15 mL of anhydrous methanol at 0 °C in a dropwise manner, according to the reported literature [36]. The resulting mixture was left at room temperature for 16 h with slow stirring. The round bottom flask was covered with a septum containing an empty balloon to control the internal pressure of the HCl gas which was evolved during the reaction. After 16 h, riboflavin (3.19 g, 8.50 mmol) was added, and the resulting solution was stirred at RT for a further 16 h. A dark brown, thick, oily syrup was formed, which was diluted with 150 mL of water and then heated at 85–90 °C for 30 minutes. Upon cooling to room temperature with stirring, orange colored FMN (**32**) precipitated (yield 67%) [45]. ^31^P-NMR (162 MHz, D_2_O), δ, ppm: −0.612; MS(ES): *m*/*z* 456.71 [M + H]^+^.

### 3.2. General Synthetic Procedure for the Preparation of Monophospho-Imidazolates (**16–19** & **29**) by Mechanochemical Ball-Milling

Activation of monophosphates (**11**–**13**, **15** & **28**) by using 1,1’-carbonyldiimidazole (CDI): A mixture of monophosphate (1 eq.), 1,1’-carbonyldiimidazole (CDI) (4 eq.) and anhydrous acetonitrile (0.3 µL/mg of monophosphate) together with 1 stainless steel ball (20 mm in diameter) was introduced into a stainless-steel jar (1.5 mL). The reaction vessel, along with another identical empty vessel, was closed and fixed on the vibration arms of a Retsch MM400 miller and vibrated at 30 Hz at room temperature for one hour. The reaction progress was monitored by ^31^P-NMR analysis (Table 2). After completion of the reaction, the resulting mixture was used for the next step without purification. Among the series of monophosphates (**11**–**13**, **15** & **28**), only compound **32** was unreactive under these conditions [36]. As an example, the sequence implemented for thiamine is presented Figure 2.

Adenosine monophosphate imidazolate (AMP-IM) (**16**): ^31^P-NMR (162 MHz, D_2_O), δ, ppm: −8.42.

Nicotinamide mononucleotide imidazolate (NMN-IM) (**17**): ^31^P-NMR (162 MHz, D_2_O), δ, ppm: −9.59.

4PYR-MP imidazolate (4PYRMP-IM) (**18**): ^31^P-NMR (162 MHz, D_2_O), δ, ppm: −8.50.

Thiamine phosphoimidazolate ThP-IM (**29**): ^31^P-NMR (162 MHz, D_2_O), δ, ppm: −8.52.

### 3.3. General Synthetic Procedure for the Preparation of Pyrophosphates (**20**–**26**, **30a**, **31**, **33** & **37**) by Mechanochemical Ball-Milling

The metallic jar was charged with monophosphate imidazolate (0.144 mmol, 1 eq.). The coupling monophosphate (0.144 mmol, 1 eq.) was added, along with MgCl_2_.(H_2_O)_6_ (0.216 mmol, 43.91 mg, 1.5 eq.), tetrazole (0.288 mmol, 20.17 mg, 2 eq.), H_2_O (0.864 mmol, 15.6 µL, 6 eq.) and a 20 mm stainless steel ball. The ball mill was set to vibrate at a frequency of 30 Hz for 90 minutes [3], after which time the jar was left to cool down to room temperature. Once the jar was opened, the product was dissolved in water, adsorbed on silica gel, and purified. Automated flash column chromatography was performed using a Teledyne ISCO CombiFlash Companion system with a RediSepRf reverse-phase C18 column (cv = 133 mL), packed with 130 g silicagel (average particle size: 40–63 microns, 230–400 mesh, 60 A^o^ pore size ) by using 100 mM ammonium acetate: acetonitrile (98%:2% to 85%:15%) as an eluent with a flow rate of 5 mL/min [27,36].

β-Nicotinamide adenine dinucleotide, NH_4_^+^ salt (β-NAD^+^) (**20**): ^1^H-NMR (400 MHz, D_2_O), δ, ppm: 9.18 (s, 1H, H-2_NAM_), 8.99 (d, *J* = 6.24 Hz, 1H, H-4_NAM_), 8.68 (d, *J* = 8.16 Hz, 1H, H-6_NAM_), 8.26 (s, 1H, H-8_Adenine_), 8.04 (t, *J* = 7.2 Hz, 1 H, H-5_NAM_), 8.00 (s, 1 H, H-2_Adenine_), 5.93 (d, *J* = 5.32 Hz, 1H, H-1_NR-ribose_), 5.87 (d, *J* = 5.92 Hz, 1H, H-1_AD-ribose_), 4.38–4.06 [m, 10H (H-2, H-3, H-4 & H-5_AB_)_NR&AD-ribose_ ]; ^31^P-NMR (162 MHz, D_2_O), δ, ppm: −11.30 & −11.64 (2P, AB_q_*, J* = 20.4 Hz & 19.4 Hz).

P_1_,P_2_-di(adenosin-5’-yl)diphosphate-bis(ammonium) salt (Ap_2_A) (**21**): ^1^H-NMR (400 MHz, D_2_O), δ, ppm: 8.11 (s, 1H, H-8_adenine_), 7.97 (s, 2H, H-2 _adenine_), 5.88 (d, *J* = 5.16 Hz, 2H, H-1_ribose_), 4.49 (t, *J* = 5.10 Hz, 2H, H-2_ribose_), 4.37 (t, *J* = 4.52 Hz, 2H, H-3_ribose_), 4.29–4.15 [m, 6H (H-4 & 2x H-5_AB_)_ribose_ ]; ^13^C-NMR (100 MHz, D_2_O), δ, ppm: 154.66, 152.15, 148.21, 139.21, 118.91 (Ar-C), 87.06 (C-1_ribose_), 83.38 (C-4_ribose_), 74.13 (C-2_ribose_), 69.99 (C-3_ribose_), 65.13 (C-5_ribose_); ^31^P-NMR (162 MHz, D_2_O), δ, ppm: −11.23.

Double NMN, NH_4_^+^ salt (**22**): ^1^H-NMR (400 MHz, D_2_O), δ, ppm: 9.28 (s, 1H, Ar-H), 9.13 (d, *J* = 6.12 Hz, 1H, Ar-H), 8.82 (d, *J* = 7.8 Hz, 1H, Ar-H), 8.14 (t, *J* = 7.2 Hz, 1H, Ar-H), 6.05 (d, *J* = 5.32 Hz, 2H, H-1_ribose_), rest of signals of d-ribose (H-2, H-3, H-4, H-5_AB_) are hidden under the broad HOD signal at 4.65; ^31^P-NMR (162 MHz, D_2_O), δ, ppm: −11.71; HRMS calcd. for C_22_H_29_N_4_O_15_P_2_[M]^+^ 651.1104 found 651.1090.

4-ox-NAD, NH_4_^+^ salt (**23**): ^1^H-NMR (400 MHz, D_2_O), δ, ppm: 8.33 (s, 1H, H-2_4py_), 8.25 (s, 2H, H-8_adenine_), 8.04 (s, 1H, H-2 _adenine_), 7.79 (dd, *J* = 2.4 Hz & 2.0 Hz, 1H, H-6_4py_), 6.39 (d, *J* = 7.52 Hz, 1H, H-5_4py_), 5.93 (d, *J* = 5.72 Hz, 1H, H-1_AD-ribose_), 5.40 (d, *J* = 6.4 Hz, 1H, H-1_4py-ribose_), 4.59 (t, *J* = 5.62 Hz, 1H, H-2_AD-ribose_), 4.38 (t, *J* = 4.46 Hz, 1H, H-2_NR-ribose_), 4.26–4.10 [m,8H (2× H-3, 2× H-4 & 2× H-5_AB_)_ribose_ ]; ^13^C-NMR (100 MHz, D_2_O), δ, ppm: 178.47 (CO-C4_4py_), 167.33 (CONH_2_), 154.64, 151.93, 148.66, 143.33, 139.67, 136.88, 120.73, 118.18, 117.24 (Ar-C), 97.0, 86.84, 83.61, 75.33, 74.32, 70.32 (d, *J*_CP_ = 12.0 Hz), 65.73 (d, *J*_CP_ = 13.6 Hz; [C1-5]ribose_(Adenosine&NR)_); ^31^P-NMR (162 MHz, D_2_O), δ, ppm: −11.30; HRMS calcd. for C_21_H_28_N_7_O_15_P_2_[M]^+^ 680.1113 found 680.1096.

Double 4PYR-MP, NH_4_^+^ salt (**24**): ^1^H-NMR (400 MHz, D_2_O), δ, ppm: 8.30 (s, 1H, H-2_4py_), 7.80 (dd, *J* = 2.24 Hz & 2.24 Hz, 2H, H-6_4py_), 6.41(d, *J* = 7.6 Hz, 2H, H-5_4py_), 5.36 (d, *J* = 6.04 Hz, 2H, H-1_ribose_), 4.16–3.95 [m, 10H (2× H-2, 2× H-3, 2× H-4 & 2× H-5_AB_)_ribose_]; ^13^C-NMR (100 MHz, D_2_O), δ, ppm: 178.47 (CO-C4_4py_), 167.31 (CONH_2_), 143.84, 137.84, 120.88, 118.85, 117.55 (Ar-C), 97.1 (C-1_ribose_), 86.84 (m,C-4 _ribose_), 75.13 (C-2_ribose_), 70.27 (C-3_ribose_), 65.39 (C-5_ribose_); ^31^P-NMR (162 MHz, D_2_O), δ, ppm: −11.32; HRMS calcd. for C_22_H_29_N_4_O_17_P_2_[M + H]^+^ 680.1113 found 680.0986.

Adenosine thiamine diphosphate (AThDP) (**30a**): ^1^H-NMR (400 MHz, D_2_O), δ, ppm: 8.32 (s, 1H, H-2_adenine_), 8.05 (s, 1H, H-8_adenine_), 7.78 (s, 1H, H-2_pyrimidine_), 5.94 (d, *J* = 6.0 Hz, 1H, H-1_ribose_), 5.24 (brs, 2H, -CH_2_N), 4.64 (brs, 2H, H-2&H-4), 4.37-4.34 (m, 1H, H-3_ribose_), 4.24-4.02 (m, 4H,H-5_A&B_, CH_2_), 3.09 (t, *J* = 5.26 Hz, 2H, CH_2_), 2.38 (s, 6H, 2× CH_3_); ^13^C-NMR (100 MHz, D_2_O), δ, ppm: 165.60, 162.15, 155.06, 152.43, 149.83, 148.92, 143.20, 139.89, 135.43, 118.87, 118.29, 105.02 (Ar-C), 86.67 (C-1_ribose_), 83.84 (m,C-4 _ribose_), 73.81 (C-2_ribose_), 70.26 (C-3_ribose_), 65.33 (C-5_ribose_), 64.73 (CH_2_), 50.25 (CH_2_), 27.43 (CH_2_), 11.08 (2× CH_3_); ^31^P-NMR (162 MHz, D_2_O), δ, ppm: −11.59; HRMS calcd. for C_22_H_30_N_9_O_10_P_2_S[M]^+^ 674.1311 found 674.1296.

Nicotinamide riboside thiamine diphosphate (NRThDP) (**31**): ^1^H-NMR (400 MHz, D_2_O), δ, ppm: 9.28 (d, *J* = 1H, H-2_NAM_), 9.13 (d, *J* = 6.32 Hz, 1H, H-4_NAM_), 8.81 (d, *J* = 8.12 Hz, 1H, H-6_NAM_), 8.14 (t, *J* = 7.17 Hz, 1H, H-5_NAM_), 7.87 (s, 1H, H-6’(Th)), 6.04 (d, *J* = 5.36 Hz, 1H, H-1_ribose_), 5.27 (br.s, 2H, -CH_2_N), 4.45–3.99 (m, 7H, H-2, H-3, H-4, H-5_A&B_ & CH_2_CH_2_P), 3.13 (t, *J* = 5.54 Hz, 2H, CH_2_CH_2_P), 2.39 (s, 3H, CH_3_), 2.33 (s, 3H, CH_3_); ^13^C-NMR (100 MHz, D_2_O), δ, ppm: 167.63 (CONH_2_), 165.60, 162.03, 154.36, 145.96, 143.14, 142.52, 139.86, 135.16, 133.83, 128.57, 118.94, 104.66 (Ar-C), 99.84 (C-1_ribose_), 86.97 (d, *J* = 7.8 Hz, C-4 _ribose_), 77.47 (C-2_ribose_), 70.62 (C-3_ribose_), 64.80-64.66 (m, 2C, CH_2_ & C-5_ribose_), 50.72 (CH_2_), 27.43 (d, *J* = 6.58 Hz, CH_2_), 11.08 (2× CH_3_); ^31^P-NMR (162 MHz, D_2_O), δ, ppm: −11.68, 11.86 (2P. AB_q_, *J* = 20.3 & 20.9 Hz); HRMS calcd. for C_23_H_31_N_6_O_11_P_2_S[M]^+^ 661.1241 found 661.1234. 

Adenosine thiamine triphosphate (AThTP) (**37**): ^1^H-NMR (400 MHz, D_2_O), δ, ppm: 8.39 (s, 1H, H-2_adenine_), 8.12 (s, 1H, H-8_adenine_), 7.79 (s, 1H, H-2 _Thiazole (Th)_), 5.99 (d, *J* = 5.96 Hz, 1H, H-1_ribose_), 5.27 (brs, 2H, -CH_2_N), 4.42–4.40 (m, 1H, H-3_ribose_), 4.26 (brs,1H,H-4_ribose_), 4.10–4.06 (m, 4H, H-5_A&B_, CH_2_), 3.13 (brs, 2H, CH_2_), 2.41 (s, 3H,CH_3_), 2.38 (s, 3H, CH_3_); ^13^C-NMR (100 MHz, D_2_O), δ, ppm: 179.96, 162.11, 155.12, 152.40, 148.87, 143.25, 135.43, 118.86, 105.26 (Ar-C), 86.61 (C-1_ribose_), 83.97 (C-4 _ribose_), 74.10 (C-2_ribose_), 70.26 (C-3_ribose_), 64.75 (C-5_ribose_), 64.73 (CH_2_), 50.20 (CH_2_), 27.49 (CH_2_), 11.11 (CH_3_); ^31^P-NMR (162 MHz, D_2_O), δ, ppm: −11.49 (1P, d, *J* = 18.54 Hz), −11.73 (1P, d, *J* = 19.37 Hz), 23.29 (1P, t, *J* = 18.74 Hz); HRMS calcd. for C_22_H_30_N_9_O_10_P_2_S[M]^+^ 674.1311 found 674.1296.

Nicotinamide riboside riboflavin diphosphate (NMN-FMN) (**33**): ^1^H-NMR (400 MHz, D_2_O), δ, ppm: 9.31 (s, 1H, H-2_NAM_), 9.19 (d, *J* = 5.08 Hz, 1H, H-4_NAM_), 8.83 (d, *J* = 8.12 Hz, 1H, H-6_NAM_), 8.20 (t, *J* = 7.14 Hz, 1H, H-5_NAM_), 7.61 (s, 1H, H-6_riboflavin_), 7.36 (s, 1H, H-9_riboflavin_), 6.07 (d, *J* = 5.4Hz, 1H, H-1_ribose_), 4.58–3.85 (m, 12H, ribose_(NR &riboflavin)_), 2.36 (s, 3H, CH_3_), 2.24 (s, 3H, CH_3_); ^13^C-NMR (100 MHz, D_2_O), δ, ppm: 179.47 (CO_NR_), 160.89 (CO_riboflavin_), 157.65 (CO_riboflavin_), 150.53, 149.78, 146.01, 142.56, 139.83, 139.35, 134.22, 133.83, 131.57, 131.44, 130.27, 128.64, 116.89, 116.79 (Ar-C), 99.91 (C-1_NR-ribose_), 87.15(C-4_NR-ribose_), 77.59 (C-2_NR-ribose_), 75.32, 72.37 (2C, CH_2riboflavin ribose_), 70.81 (C-3_NR-ribose_), 66.44 (1C, CH_2riboflavin ribose_), 64.94 (C-5_NR-ribose_), 47.49 (1C, CH_2riboflavin ribose_), 20.71 (CH_3_), 18.57 (CH_3_); ^31^P-NMR (162 MHz, D_2_O), δ, ppm: −10.67, −11.63 (2P. AB_q_, *J* = 21.5 & 20.2 Hz); HRMS calcd. for C_28_H_35_N_6_O_16_P_2_[M]^+^ 773.1579 found 773.1567.

Cyclic FMN (**34**): ^1^H-NMR (400 MHz, D_2_O): δ 7.73 (s, 1H), 7.61 (s, 1H), 5.07 (dd, *J* = 12 Hz, *J* = 16 Hz, 1H), 4.75 (d, *J* = 16 Hz, 1H), 4.71 (m, 1H), 4.47 (m, 1H), 4.32 (m, 2H), 4.13 (m, 1H), 2.53 (s, 3H), 2.39 (s, 3H). ^13^C-NMR (100 MHz, D_2_O): δ 162.40, 159.22, 152.17, 151.32, 140.83, 135.72, 135.32, 132.94, 13.78, 118.26, 76.80, 74.03, 70.83, 67.89, 49.07, 22.22, 20.04. ^31^P-NMR (162 MHz, D_2_O): δ 17.96. 

### 3.4. Synthesis of M+4 NAD

Synthesis of 1-*O*-methyl-2,3-*O*-isopropylidene-β-d-ribofuranose (**35**): To a suspension of d-ribose (**38**) (4.00 g, 26.65 mmol) in 40 mL of acetone: MeOH (1:1) mixture was cautiously added, H_2_SO_4_ (2.00 mL) and the resulting mixture was stirred at room temperature for 48 h. The reaction progress was monitored by TLC analysis. Upon completion, the reaction mixture was neutralized by the addition of solid NaHCO_3_. Once the reaction mixture reached a pH of between 6 and 7 (pH paper’s color shifts from pink to yellow) the addition of NaHCO_3_ was stopped and the resulting mixture was concentrated on the rotatory evaporator with no heating to get the desired product as a colorless oil. ^1^H-NMR in CDCl_3_ showed that the product was sufficiently pure to be used as in the next step. Yield 84%; ^1^H-NMR (400 MHz, CDCl_3_), δ, ppm: 4.95 (s, 1H, H-1_ribose_), 4.81 (d, *J* = 5.96 Hz, 1H, H-2_ribose_), 4.56 (d, *J* = 5.96 Hz, 1H, H-3_ribose_), 4.41 (brs, 1H, H-4_ribose_), 3.67 & 3.59 (AB part of ABX system, 2H, *J_AB_* = 12.5, *J_AX_* = 2.32, *J_BX_* = 3.44, H_5A_ & H_5B_), 3.41 (s, 3H, -OCH_3_), 1.46 (s, 3H,-CH_3_), 1.30 (s, 3H, -CH_3_); ^13^C NMR (100 MHz, CDCl_3_), δ, ppm: 112.07 [(CH_3_)_2_-C], 109.94 (C-1_ribose_), 88.31 (C-4_ribose_), 85.71 (C-2_ribose_), 81.44 (C-3_ribose_), 63.95 (C-5_ribose_), 26.30 (CH_3_), 24.65 (CH_3_).

Synthesis of methyl 2,3-*O*-isopropylidene-β-d-ribo-pentodialdo-1,4-furanoside (**40**): To a solution of oxalyl chloride (2.15 g, 1.45 mL, 17.00 mmol) in 15 mL, anhydrous dichloromethane was added 2 mL DMSO, diluted with 5 mL anhydrous dichloromethane in a dropwise manner at −78 °C. The resulting solution was stirred for 15 min, and then alcohol 24 (2.00 g, 9.79 mmol) dissolved in CH_2_Cl_2_ (2 mL) was added. After 15 min of stirring at −78 °C, diisopropylethylamine (6.46 g, 8.70 mL, 50 mmol) was added dropwise to the solution. For 30 minutes, the mixture was stirred at −78 °C. After 30 min, the cooling bath was removed, and the solution was warmed up to room temperature and stir for an additional 1.5 hrs. The reaction was quenched with saturated NaHCO_3_ (25 mL) and the aqueous phase was extracted with EtOAc (3 × 40 mL). The organic layers were combined, washed with saturated brine, dried over anhydrous Na_2_SO_4_ filtered, and concentrated. The crude product was sufficiently pure to be used directly in the next step without purification. Yield 77%; ^1^H-NMR (400 MHz, CDCl_3_), δ, ppm: 9.56 (s, 1H, CHO), 5.06 (s, 1H, H-1_ribose_), 5.02 (d, *J* = 5.88Hz, 1H, H-4_ribose_), 4.48 (d, *J* = 5.88 Hz, 1H, H-3_ribose_), 4.45(s, 1H, H-2_ribose_), 3.43 (s, 3H, -OCH_3_), 1.47 (s, 3H, -CH_3_), 1.31 (s, 3H, -CH_3_); ^13^C-NMR (100 MHz, CDCl_3_), δ, ppm: 167.95 (CO), 112.68 [(CH_3_)_2_-C], 109.14 (C-1_ribose_), 89.50 (C-4_ribose_), 83.98(C-2_ribose_), 80.76(C-3_ribose_), 26.19 (CH_3_), 24.84 (CH_3_).

Synthesis of methyl (3a*R*, 4*S*, 6*S*, 6a*S*)-6-methoxy-2,2-dimethyltetrahydrofuro- [3,4-d][1,3]dioxole-4-carboxylate (**41**): To a solution of aldehyde 36 (1.53 g, 7.57 mmol) in 52 mL of methanol : water mixture (6:1), solid NaHCO_3_ (121.06 mmol, 10.17 g) was added at rt. Concentrated liquid bromine (3.99 g, 1.27 mL, 25 mmol) was added to the mixture, and the orange colored solution was stirred at 40 °C for 3 h. The reaction progress was monitored by NMR analysis. Upon completion, the reaction mixture was poured into a saturated Na_2_S_2_O_3_ solution and extracted with EtOAc (3 × 20 mL). The organic layers were combined, dried over anhydrous Na_2_SO_4_ and filtered. The filtrate was concentrated, and the residue was purified by silica gel column chromatography (*n*-hexane: EtOAc = 15:1 to 2:1). Yield 51%; ^1^H-NMR (400 MHz, CDCl_3_), δ, ppm: 5.20 (d, *J* = 5.84 Hz, 1H, H-4_ribose_), 5.01 (s, 1H, H-1_ribose_), 4.59 (s, 1H, H-2_ribose_), 4.52 (d, *J* = 5.84 Hz, 1H, H-3_ribose_), 3.74 (s, 3H, -OCH_3_), 1.46 (s, 3H, -CH_3_), 1.31 (s, 3H, -CH_3_); ^13^C-NMR (100 MHz, CDCl_3_), δ, ppm: 170.57 (CO), 112.69 [(CH_3_)_2_-C], 109.28 (C-1_ribose_), 84.28 (C-4_ribose_), 83.46 (C-2_ribose_), 82.07 (C-3_ribose_), 55.26 (-OCH_3_), 52.16 (C-5_ribose_), 26.32 (CH_3_), 24.94 (CH_3_).

Synthesis of methyl-2,3-*O*-isopropylidene-β-d-ribofuranoside[D2] (**42**): 

(i) Method A: LiAlH_4_ (0.07 g, 1.60 mmol) was added to a solution of compound 41 (1.42 g, 6.11 mmol) in 10 mL of anhydrous diethyl ether. The resulting mixture was stirred at room temperature for 36 h, and the reaction progression was monitored by TLC analysis. Upon completion, the resulting mixture was passed through a celite pad and concentrated. The crude mixture was purified by silica gel column chromatography (n-hexane: EtOAc = 2:1 to 0:1), Yield 35%. 

(ii) Method B: Compound 41 (3.50 g, 15.08 mmol) was suspended in 50 mL of D_2_O (d-Enrichment > = 99.95%, ACROS ORGANICS) and then 662 mg (15.8 mmol) of NaBD_4_ was added in a portion-wise manner with stirring. Upon the complete addition of NaBD_4_, the resulting mixture was stirred at rt for 24 h with an empty balloon to control the internal pressure of deuterated hydrogen in a round bottom flask. After 24 h, ^1^H-NMR of crude showed complete conversion of starting to the product. Upon completion, the reaction mixture was stirred with EtOAc (2 × 25 mL) for 4–5 h, and the separated organic layers were combined, dried over anhydrous Na_2_SO_4_, and concentrated. The residual product was of sufficient purity to be used directly for the next step without purification. Yield 64%.; ^1^H-NMR (400 MHz, CDCl_3_), δ, ppm: 4.95 (s, 1H, H-1_ribose_), 4.81 (d, *J* = 5.92 Hz, 1H, H-2_ribose_), 4.57 (d, *J* = 5.92 Hz, 1H, H-3_ribose_), 4.40 (s, 1H, H-4_ribose_), 3.42 (s, 3H, -OCH_3_), 1.47 (s, 3H, -CH_3_), 1.30 (s, 3H, -CH_3_); ^13^C-NMR (100 MHz, CDCl_3_), δ, ppm: 112.08 [(CH_3_)_2_-C], 109.96 (C-1_ribose_), 88.21 (C-4_ribose_), 85.79 (C-2_ribose_), 81.42 (C-3_ribose_), 66.48–63.05 (m, C-5_ribose_), 55.48 (-OCH_3_), 52.16 (C-5_ribose_), 26.31 (CH_3_), 24.66 (CH_3_).

Synthesis of C-5-deuterated ribose tetraacetate (**46**): (a) Step-1: A solution of methyl-2,3-*O*-isopropylidene-β-d-ribofuranoside [D2] (**42**)(1.00 g, 4.03 mmol) in a 5 mL mixture of TFA: H_2_O (1.3:1) was stirred at room temperature until the acetonide was completely removed. The reaction progression was monitored by ^1^H-NMR analysis. Upon completion, the reaction mixture was concentrated under reduced pressure, and the crude oily residue was used in the next step.

Step 2: 0.61 g of step 1 product was dissolved in 10 mL of anhydrous MeOH, followed by 60 µL of conc. H_2_SO_4_.The resulting mixture was stirred at RT overnight in a nitrogen environment. The reaction progression was monitored by ^1^H-NMR analysis. Upon completion, the reaction mixture was neutralized by adding pyridine. Once the reaction mixture reached a pH between 6 and 7, the resulting mixture was concentrated on the rotatory evaporator to get the reaction intermediate 44, which was used directly in the next step without purification.

Step-3: The crude intermediate generated in step-2 (0.67 g) was dissolved in 5 mL of anhydrous pyridine, and then 1.5 mL of acetic anhydride was added. The resulting mixture was stirred at room temperature overnight under nitrogen. The reaction progress was monitored by ^1^H-NMR analysis. Upon completion, the solvent was removed under reduced pressure to obtain the crude product that was dissolved in 15 mL of DCM and stirred with 30 mL of aqueous copper sulfate solution (15%) until all the residual pyridine was completely removed. Upon completion, the organic layer was separated and dried over anhydrous Na_2_SO_4_, concentrated. The crude (**45**) was used in the next step.

Step-4: A solution of crude intermediate generated in step-3 (1.00 g) in 10 mL of acetic acid was added slowly to acetic anhydride (3.23 mL, 34.16 mmol) at 0 °C and the reaction mixture was stirred for 15 min. The 0.5 mL of conc. H_2_SO_4_ was then added in a dropwise manner with stirring. Upon completion of the addition, the ice bath was removed, and the resulting mixture was stirred at room temperature for 3 h. It is important to note that longer stirring time facilitates the formation of the pyranose form, yielding a mixture of pyranose and furanose forms of C-5 deuterated ribose tetraacetate. After 3 h, a ^1^H-NMR analysis of the crude showed that the desired furanoside had formed preferentially. Crushed ice was then added, and the quenching reaction mixture was stirred for 1 hour to allow for the complete hydrolysis of excess Ac_2_O to acetic acid. After that time, 15 mL of CHCl_3_ was added to the mixture, which was stirred for 30 minutes (repeated 3 times). Upon extraction with ethyl acetate, the organic layers were combined and stirred with 50 mL of a saturated NaHCO_3_ solution for 1 hour and then separated. The organic layer was dried over anhydrous Na_2_SO_4_, concentrated under a reduced vacuum to afford the desired product, 46 as an α/β mixture. The product was used in the next step without purification. Yield 69%. ^1^H-NMR (400 MHz, CDCl_3_), δ, ppm: 6.35 (d, *J* = 4.16 Hz, 1H, H-1α_ribose_), 6.09 (s, 1H, H-1β_ribose_), 5.29–5.26 (m, 2H, H2&3β_ribose_), 5.17–5.15(m, 2H, H2&3α_ribose_), 2.15–2.00 (m, 18H, 6xOAc_α/β mixture_); ^13^C-NMR (100 MHz, CDCl_3_), δ, ppm: 170.42, 170.08, 169.64, 169.37, 169.31, 168.95, 166.36 (8× OCOCH_3 α/β mixture_), 98.16, 94.00, 81.47, 79.16, 74.10, 70.49, 69.96, 69.70 (C-1,C-2,C-3,C-4,C-5 _ribose, α/β mixture_), 20.98, 20.71, 20.68, 20.56, 20.44, 20.40, 20.25 (8× OCOCH_3 α/β mixture_).

Synthesis of C-5 deuterated adenosine (**48**):

Step-1: A 25 cm^3^ PTFE ball milling vessel was charged with crude C-5 deuterated ribose tetraacetate sugar 46 (0.30 g, 0.94 mmol), TMSOTf (0.42 g, 1.87 mmol), adenine (0.13 g, 0.94 mmol) and a PTFE ball bearing. The vessel was shaken with a Retsch MM 400 mixer mill for 35 min at 30 Hz. The crude material was dissolved in a mixture of MeOH: DCM (1:1), adsorbed on silica gel, and purified by flash column chromatography using n-hexane: DCM (1:4 to 0:1). After purification, an α/β mixture of the triacetylated nucleoside intermediate (**47**) was isolated.

Step 2: The intermediate isolated in step 1 (0.37 g, 0.94 mmol) was dissolved in a 9 mL mixture of MeOH and NH_4_OH (2:1) and stored in the refrigerator at 8 °C for 24 h. After that time, a ^1^H-NMR of the crude was recorded in D_2_O to ensure that the complete removal of all three acetates had occurred. Upon completion, the solvent was removed under reduced pressure and the crude was treated with 2 mL of MeOH and kept at −20 °C for a week to precipitate the β-isomer as a white powder (**48**). Yield 25% (β-isomer) The supernatant was concentrated under reduced pressure and stored for future use. ^1^H-NMR (400 MHz, D_2_O), δ, ppm: 8.18 (s, 1H, H-2_adenine_), 8.08 (s, 1H, H-8_adenine_), 5.92 (brs, 1H, H-1_ribose_), 4.31 (s, 1H, H-4_ribose_), 4.18 (s, 1H, H-3_ribose_); ^13^C-NMR (100 MHz, DMSO-d^6^), d, ppm: 156.57 (d, *J* = 6.48 Hz), 152.82, 149.51, 140.38 (Ar-C), 88.32 (d, *J* = 3.24 Hz, C-1_ribose_), 86.19 (C-4_ribose_), 73.80 (d, *J* = 10.82 Hz, C-2_ribose_), 71.02 (d, *J* = 10.39 Hz, C-2_ribose_), 62.00-61.28 (m, C-5_ribose_); HRMS, *m*/*z*: found 270.1165 [M + 1], calculated for C_10_H_11_D_2_N_5_O_4_ 270.1157 [M + 1].

Synthesis of C-5 deuterated adenosine monophosphate triethylamine salt (**49**): Using a heat gun, C-5 bis-deuterated adenosine (**48**) (0.07 g, 0.26 mmol) was dissolved in 1 mL of (MeO)_3_PO. The resulting solution was cooled and stirred at −20 °C for 20 min. POCl_3_ (72 µL) was added. The resulting mixture was stirred at −20 °C for a few min more before being stored at −20 °C in a refrigerator for 24 h, after which time 50 µL of the crude was dissolved in 450 µL of D_2_O and stirred at RT for 1 hour before being analyzed by ^1^H- and ^31^P-NMR to ensure complete phosphorylation of adenosine 48. Upon confirmation, 1.5 mL of water was added to the reaction mixture. The resulting solution was stirred for 2 h at RT. The pH of the solution was subsequently adjusted to 7 by adding Et_3_N. The solution was freeze-dried to afford the triethylammonium salt of C-5 bis-deuterated adenosine monophosphate (**49**) as a white solid. ^1^H- and ^31^P-NMR showed some amount of unreacted M+2 adenosine (8%) with C-5 bis-deuterated adenosine monophosphate (**49**). The crude material was used directly for the next step without further purification. Yield 80% (49, β-isomer); ^1^H-NMR (400 MHz, D_2_O), δ, ppm: 8.43 (s, 1H, Ar-H), 8.21 (s, 1H, Ar-H), 6.04 (d, *J* = 5.6Hz, 1H, H-1_ribose_), 4.41 (brs, 1H, H-3_ribose_), 4.28 (brs, 1H, H-4_ribose_); ^31^P-NMR (162 MHz, D_2_O), δ, ppm: 0.107. 

Synthesis of [^18^O]Nam (**51**): Synthesized according to the reported procedure [3].

Synthesis of mono-silylated [^18^O]Nam (**52**): A 50 mL single-neck round-bottom flask was charged with 1.24 g (0.01 mol) of [^18^O] Nam (**51**), followed by the addition of 12 mL of HMDS and 2.6 mL of TMSCl (2.23 g; d = 0.856; 0.02 mol) from a syringe. The reaction mixture was immersed in an oil bath and heated at 120–130 °C with a reflux condenser under an argon atmosphere overnight with stirring. The reaction mixture was allowed to cool down to room temperature. The ^1^H-NMR (C_6_D_6_) spectrum of the reaction mixture indicated a complete transformation of the starting labeled Nam into a mono-silylated derivative. Volatiles were removed from the reaction mixture under reduced pressure to give a white solid of silyl-[^18^O]Nam (**52**), 1.90 g (96%). ^1^H-NMR (C_6_D_6_), δ, ppm: 8.82 (d, *J* = 2.0 Hz, 1H), 8.35 (dd, *J* = 4.8, 1.7 Hz, 1H), 7.72 (dt, *J* = 7.9, 2.0 Hz, 1H), 6.54 (ddd, *J* = 4.8, 7.9, 0.7 Hz, 1H), 5.19 (brs, 1H, NH), 0.16 (s, 9H, SiMe_3_).

Synthesis of [^18^O] NR TA OTf (**53**): A Retsch MM400 mill’s PTFE jar was loaded with 1.82 g (9.3 mmol) of silyl-[^18^O] Nam (**52**) 2.86 g (9.0 mmol) of β-d-ribofuranose 1,2,3,5-tetraacetate was added to the jar, followed by the addition of 0.6 mL of dry DCM and TMSOTf (2.36 g; 10.6 mmol; 1.2 eq.). The jar was mounted on the ball miller, and the reaction mixture was shaken at 30 Hz for 30 min. The jar was cooled down to room temperature. The content of the jar was dissolved in dry DCM. The solution was transferred into a round-bottom flask and the solvents were removed under reduced pressure to yield 5.80 g of a yellow foam of [^18^O]NR TA-OTf (**53**)(96% yield).^1^H-NMR (D_2_O), δ, ppm: 9.37 (s, 1H, H2), 9.13 (d, 1H, *J* = 6.4 Hz, H6), 8.92 (d, 1H, *J* = 8.0 Hz, H4), 8.21 (dd, 1H, *J* = 6.5 Hz, *J* = 8.7 Hz, H5), 6.51 (d, 1H, *J* = 3.7 Hz, H1′), 5.49 (dd, 1H, *J* = 3.8 Hz, *J* = 5.8 Hz, H2′), 5.38 (apparent t, 1H, *J* = 5.4 Hz, H3′), 4.80–4.83 (m, 1H, H4′), 4.42–4.50 (m, 2H, H5′), 2.09 (s, 3H, Me), 2.06 (s, 3H, Me), 2.02 (s, 3H, Me).

Synthesis of [^18^O] NR OTf (**54**): In a 130 mL pressure tube closed with a septum, a solution of [^18^O] NR TA OTf (**53**) (2.44 g, 4.00 mmol) in anhydrous methanol (30 mL) was prepared. The solution was cooled down to −78 °C, and ammonia gas (passed through a tube filled with NaOH) was bubbled into the solution for 5 minutes while stirring. The reaction solution was additionally stirred at −78°C for 10 min, the septum was removed, and the tube was immediately closed with a PTFE cap and transferred into a freezer and kept there at −20 °C for 6 days. The pressure tube was transferred into an ice bath (0 °C) and the contents of the tube were transferred via a cannula into a recovery flask kept at 0 °C. The recovery flask was attached to a rotary evaporator, and ammonia gas was evaporated without any external heating, which resulted in the continuous maintenance of the evaporated solution temperature below 0 °C. After complete removal of ammonia, any residual methanol was removed at 23 °C and the oily residue was kept under a high vacuum to give a viscous yellow liquid (2.05 g, 90% as calculated based on a mixture of [^18^O] NR OTf and acetamide by-product). ^1^H-NMR (D_2_O), δ, ppm: 9.48 (s, 1H, H2), 9.15 (d, 1H, *J* = 6.2 Hz, H6), 8.85 (d, 1H, *J* = 8.1 Hz, H4), 8.16 (dd, 1H, *J* = 6.5 Hz, *J* = 8.7 Hz, H5), 6.12 (d, 1H, *J* = 4.4 Hz, H1′), 4.38 (t, 1H, *J* = 4.8 Hz, H2′), 4.35–4.36 (m, 1H, H4′), 4.23 (t, 1H, *J* = 4.6 Hz, H3′), 3.78 and 3.93 (AB part of ABX system, 2H, *J_AB_* = 12.9 Hz, *J_AX_* = 3.5 Hz, *J_BX_* = 2.9 Hz, H5′_A_ and H5′_B_), 3.21 (s, 0.8H, Me of CH_3_OH), 1.86 (s, 8H, Me of CH_3_CONH_2_). MS: found *m*/*z* = 257.10 (M-OTf). ^19^F-NMR (D_2_O), δ, ppm: −78.81.

Synthesis of [^18^O] NMN (**15**): In a pressure tube closed with a septum, a solution of [^18^O] NR OTf (**54**) (2.05 g, with acetamide admixture; ca. 4.50 mmol) in trimethyl phosphate (18 mL) was prepared. The solution was cooled down in an ice/NaCl bath to −15 °C, and phosphorus oxychloride (2.8 g; 0.018 mol, ca. 4 eq.) was added from a syringe with stirring. The septum was removed, and the tube was immediately closed with a PTFE cap, transferred into a freezer, and kept there at −20 °C for 5 days. The ^1^H-NMR (D_2_O) spectrum of the reaction mixture aliquot indicated the presence of a characteristic splitting pattern of the P-O-CH_2_-CH moiety and the consumption of all the starting [^18^O] NR (**54**). A mixture of 60 mL of anhydrous diethyl ether and 10 mL of anhydrous ACN was added to the reaction solution. The resulting solution was poured into 250 mL of anhydrous diethyl ether in an Erlenmeyer flask. The mixture was left for 2 h at room temperature to allow the forming yellowish precipitate to completely settle out. The supernatant was removed by decantation, and the precipitate was washed with anhydrous diethyl ether (3 × 25 mL). Water (15 mL) was added to the precipitate in the flask. The reaction solution was stirred at room temperature for 20 min. Volatiles were removed under reduced pressure to give a viscous residue. The residue was applied on DOWEX resin column (L = 30 cm, D = 2 cm); eluent: water to a 1:10 formic acid/water mixture. The fractions were monitored by ^31^P/^1^H-NMR spectroscopy, and those containing the desired product were collected, combined, and evaporated to give the desired compound (0.74 g, 54%). ^1^H- and ^31^P-NMR showed some percentage of unreacted ^18^O-NR with ^18^O-NMN.The product was used directly for the next step without further purification. ^1^H-NMR (400 MHz, D_2_O), δ, ppm: 9.40 (s,1H,Ar-H), 9.20 (d, *J* = 6.28 Hz, 1H, Ar-H), 8.88 (d, *J* = 8.16 Hz, 1H, Ar-H), 8.20 (t, *J* = 7.16 Hz, 1H, Ar-H), 6.10 (d, *J* = 5.52 Hz, 1H, H-1_ribose_), 4.53–4.34 (m,2H,H-2&4_ribose_), 4.36–4.34 (m, 1H, H-3_ribose_), 4.16 & 4.00 (AB part of ABX system, 2H*, J_AB_* = 12.00, *J_AX_* = 5.56, *J_BX_* = 4.76, H_5A_ & H_5B_); ^13^C-NMR (100 MHz, D_2_O), δ, ppm: 165.78 (CO), 146.08, 142.72, 139.79, 133.93, 128.49 (Ar-C), 100.12 (C-1_ribose_), 87.69 (d, *J* = 8.7 Hz, C-4_ribose_), 77.71 (C-2_ribose_), 71.12 (C-3_ribose_), 63.72 (d, *J* = 4.63 Hz,C-5_ribose_); ^31^P-NMR (162 MHz, D_2_O), δ, ppm: 1.12.

Synthesis of [^18^O] NMN imidazolate (**19**): Described in section B. ^31^P-NMR (162 MHz, D_2_O), δ, ppm: −9.68.

Synthesis of [^18^O]M+4NAD and [^18^O]M+4 NMN (**25** & **26**) as described above:

[^18^O] β-Nicotinamide adenine dinucleotide, NH_4_^+^ salt (M+4 β-NAD^+^) (**25**): isolated yield: 8%; ^1^H-NMR (400 MHz, D_2_O), δ, ppm: 9.24 (s, 1H, H-2_NAM_), 9.06 (d, *J* = 6.52 Hz, 1H, H-4_NAM_), 8.36 (d, *J* = 8.08Hz, 1H, H-6_NAM_), 8.34 (s, 1H, H-8_Adenine_), 8.12–8.08 (m, 2H, H-2_Adenine &_ H-5_NAM_ ), 5.98 (d, *J* = 5.64 Hz, 1H, H-1_NR-ribose_), 5.95 (d, *J* = 6.0 Hz, 1H,H-1_AD-ribose_), 4.45-4.11 [m, 8H (H-2,H-3,H-4)_NR&AD-ribose_ & H-5_AB NR-ribose_)]; ^13^C-NMR (100 MHz, D_2_O), δ, ppm: 152.00, 145.6, 142.2, 139.7, 128.5, 118.9 (Ar-C), 99.9 (C-1_ribose_), 77.5 (C-2_ribose_), 73.8 (C-3_ribose_), 70.3 (d, *J* = 19.95 Hz, C-5_ribose_).^31^P-NMR (162 MHz, D_2_O), δ, ppm: −11.27 & −11.67 (2P, AB_q_, *J* = 19.94 Hz&20.78 Hz); HRMS, *m*/*z*: found 668.1331 [M + 1], calculated for C_21_H_26_D_2_N_7_^18^O^16^O_13_P_2_ 668.1304 [M + 1].

[^18^O] Double NMN, NH_4_^+^ salt (M+4 NMN) (**26**): Isolated yield:11%; ^1^H-NMR (400 MHz, D_2_O), δ, ppm: 9.31 (s, 1H, H-2_NAM_), 9.16 (d, *J* = 6.12 Hz, 1H, H-4_NAM_), 8.65 (d, *J* = 8.12 Hz, 1H, H-6_NAM_), 8.17 (t, *J* = 7.22Hz, 1H, H-5_NAM_), 6.08 (d, *J* = 7.64 Hz, 1H,H-1_ribose_), 4.50–4.10 [m, 10H, (H-2,H-3,H-4,H-5_AB_) _ribose_]; ^31^P-NMR (162 MHz, D_2_O), δ, ppm:−11.71; MS(ES): *m*/*z* 655.27 [M + H]^+^.

Spectrometric and spectroscopic analyses of all the materials described herein are available in the Appendix A section.

## 4. Conclusions

A novel two-step one pot approach to the generation of unsymmetrical canonical and non-canonical dinucleotides in isolated, UV-quantified yields that are in par and even exceed that of the current protocols of simpler dinucleotides has been demonstrated. It offers means to generate valuable, new chemical tools, at scale that enable more bold biological investigations for which milligrams rather than microgram of labeled materials of dinucleotides would be required to pursue functional investigations. Studies related to bio-distributions in animal models, transporters of extracellular and intracellular (di)nucleotides, and identifications of novel biosynthetic pathways that rely on traceable chemical tools could greatly benefit from this versatile chemistry. Crucially, the reaction time reduction and the ease of chemical handling that this approach offers, provide scope for implementation in settings seeking to perform radio-emitting isotopic labelling.

## Data Availability

Data and additional details on the synthetic procedures are available upon request.

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
