# Peer review of "Synthesis of Mixed Dinucleotides by Mechanochemistry"

_molecules, 2022, doi:10.3390/molecules27103229_

Round 1

Reviewer 1 Report

In this manuscript, the authors describe the mechanochemcial synthesis of vitamin B1, B2, and B3 derived nucleotides and dinucleotides. It was found that adenosine and thiamine were particularly amenable to milling conditions. Mixed dinucleotides of adenine and thiamine (vitamin B1), riboflavin (vitamin B2), nicotinamide riboside and 3-carboxamide 4-pyridone riboside (both vitamin B3 derivatives) could be generated under the optimized conditions for the pyrophosphate bond formation. This manuscript can be accepted for publication after the authors address the following issues.

-The compound numberings should be bolded throughout the manuscript including those in the figures and tables.

-The compounds should be numbered sequentially and mentioned in the text. The compound numberings and compound abbreviations were used and thus hard to follow and confusing to identify a specific compound in some cases. For example, it remains unclear for the identities of compounds 16-22 in Figure 2, and thus should be clarified. It is better that all compounds are numbered, and common abbreviations for some compounds can be used with the corresponding compound numberings simultaneously.

-L132-L133: “…a mixture of unreacted adenosine (74%) and product AMP (36%)” were not consistent with those shown in Table 1. Please check and correct!

-The yields shown in Table 2 were generally quite low, can the reaction conditions such as reagent ratio, reaction time, milling time, LAG agent be varied to significantly improve the product yields?

-L194: “Isolated yields ranged between 12% and 65%.” did not agree with the data shown in Table 2. Please check and correct!

-The chemical structure of the starting material 25 should be given for clarity.

-L256: The mentioned Figure 7 was missing. Strangely, there were another two Figure 7 in L397-398 and L648-649, and another Figure 6 in L499 besides L231.

-A comprehensive review on mechanochemical organic synthesis published in Soc. Rev., 2013, 42, 7668 should be cited.

-The reference citations should be consistent with the journal requirements and consistent, e.g., some references showed full journal names, while others were not; the journal name of Ref 36 was not complete….

Author Response

We thank the reviewer for their suggestions and have sought to address each point individually in our revision.  They are as follows:

-The compound numberings should be bolded throughout the manuscript including those in the figures and tables.

-The compounds should be numbered sequentially and mentioned in the text. The compound numberings and compound abbreviations were used and thus hard to follow and confusing to identify a specific compound in some cases. For example, it remains unclear for the identities of compounds 16-22 in Figure 2, and thus should be clarified. It is better that all compounds are numbered, and common abbreviations for some compounds can be used with the corresponding compound numberings simultaneously.

Reply: The manuscript has been updated. All of the numbers are now in bold font and in order. Numbers in the text correspond to compound abbreviations.

-L132-L133: “…a mixture of unreacted adenosine (74%) and product AMP (36%)” were not consistent with those shown in Table 1. Please check and correct!

Reply: Corrected now the line number is 137-138.

-The yields shown in Table 2 were generally quite low, can the reaction conditions such as reagent ratio, reaction time, milling time, LAG agent be varied to significantly improve the product yields?

Reply: Table-2 is the comparison of two reported methodologies, Method A (Reference 27) and Method B (Reference 36), in respect to their yield and the reaction conditions that were adopted, are the best in terms of symmetric and asymmetric dinucleotide syntheses. We applied these two methods to the development of a new class of dinucleotides. The yields shown in table 2 are optimized. However, it is important to differentiate between reaction and isolated yields.  Dinucleotide chemistry often fails to deliver high yields upon purifications.  We have taken great care to provide accurate isolated yields, measured on UV absorbance.  Many publications report yields based on mass of product isolated. However, these are often inflated by the presence of salts used in the purification protocols (e.g., TEAB or ammonium bicarbonate).  We have been exploring the use of boronate-resins for purification by Teledyne to improve recovery and separation but have not yet been able to optimize the process to satisfaction.   

-L194: “Isolated yields ranged between 12% and 65%.” did not agree with the data shown in Table 2. Please check and correct!

Reply: Corrected. Indeed, its 12% to 63% not 65%.

-The chemical structure of the starting material 25 should be given for clarity.

Reply: The structures of thiamine and its derivatives have been revised throughout the manuscript. Now the C=N bond is clear in all the structures.

-L256: The mentioned Figure 7 was missing. Strangely, there were another two Figure 7 in L397-398 and L648-649, and another Figure 6 in L499 besides L231.

Reply: The figures have been corrected. The figures, except for Figure 1, have been renamed “schemes” (1-7).

-A comprehensive review on mechanochemical organic synthesis published in Soc. Rev., 2013, 42, 7668 should be cited.

Reply: addressed. Line 81

-The reference citations should be consistent with the journal requirements and consistent, e.g., some references showed full journal names, while others were not; the journal name of Ref 36 was not complete….

Reply: All the typos and errors have been corrected and the full journal names have been abbreviated in reference section.

Reviewer 2 Report

Comments on the ms of Migoud et al :

  • 2 is too crowded. It should be transformed into 3 separate schemes for clarity.

–     In the first step of the reaction sequence shown in Fig. 2, a few questions emerge:

1., Why is there need for 3-4 equiv-s of POCl3?

2., Why was not the hydrolysis step shown?

3., What is the reason for performing the phosphorylation in ball and tube mill?

–     In Fig. 2, the compound numbers should be boldface.

Fig. 2/text: “solution phase” was abbreviated as “SP”, but this was not mentioned.

“CDI” should be explained (Fig. 2).

“1 drop CAN” is not a precise expression (Fig. 2).

  • In Fig. 3, there are no compound numbers.

–     Under point 2.2, there is no mention to the 3. part of Fig. 2. The pyrophosphate was not numbered. Would it be possible to insert a scheme here?

–     In Table 1, trimethyl phosphate was shown in 2 variations.

  • Table 2: “step-wise” is one word.

–     The experimental details in Table 2 are two detailed.

–     There is no reason to present separately the short “Discussion”. “Results & Discussion” would be better together.

–     In the General Procedures, please provide all g or ml quantities along with the mmols. It is not good to use “eq.”.

–     It is not nice to have Figure 7 in the Experimental.

–     By the way the Figures are mostly Schemes.

–     Would it be possible to assign the 13C NMR data?

–     Minor issues:

page 1, line 40: please elucidate “B-vitaminome”.

page 3, line 111: please write “hydroxy” instead of “hydroxyl”.

In summary, the recommendation is major revision.

Author Response

We thank the reviewer for their suggestions. We have endeavored to address each point individually:

- 2 is too crowded. It should be transformed into 3 separate schemes for clarity.

Reply: Figure/Scheme2 has been split in three parts to make it much clear and easy for the readers.

–     In the first step of the reaction sequence shown in Fig. 2, a few questions emerge:

1., Why is there need for 3-4 equiv-s of POCl3?

Reply: We optimized this reaction sequence and found that 4 eqv POCl3 offered max phosphorylation with short reaction times, except in the case of adenosine.  Unlike the nucleosides, riboflavin was completely inactive under ball-milling conditions.  Water content present in the alcohol, solvent-nucleoside activation by complexation (Yoshikawa chemistry), hydrolysis of stored POCl3 overtime and rapid under ambient humid conditions can all be reasons to drive the optimized equivalency.

2., Why was not the hydrolysis step shown?

Reply: Hydrolysis step has been added in step 1

3., What is the reason for performing the phosphorylation in ball and tube mill?

Reply: This has now been addressed in the introduction at line 79 onwards.  The example of thiamine is particularly relevant to this question since thiamine is a particularly difficult chemical to dissolve in any (aqueous or organic) solvent.

–     In Fig. 2, the compound numbers should be boldface.

Reply: Done.

Fig. 2/text: “solution phase” was abbreviated as “SP”, but this was not mentioned.

Reply: Corrected, SP has been mentioned in figure 2 (Now Scheme 1).

“CDI” should be explained (Fig. 2).

Reply: CDI has been explained in Step 1(Fig 2)

“1 drop CAN” is not a precise expression (Fig. 2).

Reply: Instead of 1 drop, we have mentioned the exact amount of acetonitrile used in the process.

In Fig. 3, there are no compound numbers.

Reply: Compounds have been numbered in Figure 3 (Now Scheme 2).

–     Under point 2.2, there is no mention to the 3. part of Fig. 2. The pyrophosphate was not numbered. Would it be possible to insert a scheme here?

Reply:  We have indicated in line 189 where the pyrophosphate chemistry could be found in scheme 1.  Moving a scheme to describe this step independently of the two others would detract from the fact that the latter part of this chemistry is a two-step one pot process.

–     In Table 1, trimethyl phosphate was shown in 2 variations.

Reply: Corrected.

Table 2: “step-wise” is one word.

Reply: Corrected.

–     The experimental details in Table 2 are two detailed.

Reply: The table 2 has been simplified

–     There is no reason to present separately the short “Discussion”. “Results & Discussion” would be better together.

Reply: Amended

–     In the General Procedures, please provide all g or ml quantities along with the mmols. It is not good to use “eq.”.

Reply: Corrected (Heading : 4.3 General synthetic procedure for the preparation of pyrophosphates (20-26, 30a,31,33 &37) by mechanochemical ball-milling).  We have made appropriate changes to reflect the reviewer’s preference. However, one key advantage to chemistry by milling is the fact that instead of using balk equivalency of solvents and sometimes reagents, one can reduce this dramatically. When we indicate milligrams or milliliter of reagents and solvent rather than equivalencies to the limiting reagent, we lose track of the importance of minimizing our synthetic atom footprint, hence the choice of reporting equivalency rather than actual amounts.

–     It is not nice to have Figure 7 in the Experimental.

Reply: If the reviewer and the editor think it is more appropriate to have this synthetic sequence in the supplementary materials, we are happy to oblige.

–     By the way the Figures are mostly Schemes.

Reply: Except Figure1, rest of the figures (2-8) have been named schemes (1-7).

–     Would it be possible to assign the 13C NMR data?

Reply: Unfortunately, we are not able to adequately assign the 13C NMR of the many compounds that we have reported in this manuscript. We appreciate the request made by the reviewer as it would indeed be valuable for future reference, although some of the reported materials are known entities and already very well characterized.  Critically, for  comprehensive and accurate 13C NMR assignments, we would have to conduct extensive 1H-13C, 13C-13C and 1H-31P 2D NMR analyses on materials that we no longer have since most are now in use in biological investigations, and would have to resynthesize. 

–     Minor issues:

page 1, line 40: please elucidate “B-vitaminome”.

Reply: Corrected

page 3, line 111: please write “hydroxy” instead of “hydroxyl”.

Reply: Corrected

Reviewer 3 Report

Figure 1 R1 and R2 should be indicate who they are

Figure 2 a contranion in the last structure is required to neutralize the N+ in thiazolium ring.

Table 2 check for some misspelling

Authors shoul difference betwee figure and scheme, they refer to several schemes as figures.

There are two figures 6 and 7.

In some figures that should be named as schemes, Authors should consider to use the caption to describe the reactions conditions instead of the arrows in the scheme.

The article is well written and the subject is of interest and fall into green chemistry approach.

The references should be as indicated in the instructions for authors

Journal Articles:
1. Author 1, A.B.; Author 2, C.D. Title of the article. Abbreviated Journal Name Year, Volume, page range

Author Response

We thank the reviewer for their suggestions.  We have tried to address each point individually as follows: 

Figure 1 R1 and R2 should be indicate who they are:  

Reply: With this figure, we have attempted to provide the most encompassing way to describe the chemistry that we have performed without creating a complex figure.  B1 and B2 can be any of the reagents indicated in the figure.  We use the “red oxygen” in the reagents and the product as we wanted to describe how diverse the pool of product could be.  A bold oxygen might be required if the publication is black only.

Figure 2 a contranion in the last structure is required to neutralize the N+ in thiazolium ring.

Reply: A chloride (Cl-) counterion has been added with a thiamine structure.

Table 2 check for some misspelling

Reply: Addressed

Authors shoul difference betwee figure and scheme, they refer to several schemes as figures. There are two figures 6 and 7. In some figures that should be named as schemes, Authors should consider to use the caption to describe the reactions conditions instead of the arrows in the scheme.

Reply: Addressed. However, we are partial to the reaction conditions being written over/under the arrows).

The article is well written and the subject is of interest and fall into green chemistry approach.

The references should be as indicated in the instructions for authors

Journal Articles:

  1. Author 1, A.B.; Author 2, C.D. Title of the article. Abbreviated Journal Name Year, Volume, page range.

Reply: We have sought to revise the reference section according to the instructions to author.  However, we are to do this manually as our endnote version does not recognize the DMPI format.  We ask the editor and the reviewer to allow us some leeway until the manuscript has been accepted in principle and the final version is being prepared for publication to have the reference section appropriately edited according to the editors' recommendations. 

Reviewer 4 Report

This is a very substantial piece of extremely useful synthetic work on very challenging systems. It was a pleasure to read through the work and I recommend 'minor corrections'. There are a few experimental details that must be tidied up, but, beyond these a some typos (listed below), this work very much deserves to be published and shared with the nucleotide community.

1) The title mentions only mechanochemistry and not the significant amount of solution-based work that is held within the manuscript.

2) page 1, line 17: "perfectly" is not a fair claim to make.

3) page 1, line 33 cofactor, not cofactors.

4) page 2, lines 82-85: the later portion of the manuscript and SI do not same to provide any substantiation of quantitative data to support degradation/chlorination outcomes, nor is literature evidence provided. Please provide supporting data.

5) The presentation of Figure 1 is a little unclear (which are R1, R2?) and a little messy (space lines at more regular angles, consistency of molecule spacings etc.

6) In Figure 1 and throughout the manuscript the thiazolium structure is not very clear—the + charge seems to be displaced such that it obscures the C=N bond to the point that it is difficult to see—please rectify.

7) page 3, line 131-133 and through the manuscript: use correct time abbreviations consistently i.e. h for hours (h is used in some places, hours elsewhere), min for minutes (mins is non-standard).

8) page 4, line 135 '...unstuitable FOR riboflavin...'

9) Figure 2: I found this (and most other figures) to be difficult to read because of the small text fonts (atom labels are ok). The table within Figure 2 is particularly hard to read. Please increase font size/consistency here and throughout the Figures, as needed.

10) Figure 3—make C=N clear in all structures.

11) Final entry of Table 1: The use of methanol is surprisingly described as NOT leading to methyl phsophoesters. Looking at the volumes of reagent/'solvent' used, this is a little surprising. Some degree of explanation and/or literature referencing, if this is a known process should be included.

12) line 168: each phosphoester (not plural)

13) line 169: synthesEs (plural)

14) line 173: 1 h, not 1 hr, and similar corrections for time units on lines 182, 185.

15) line 192: Some brief details of chromatography methodology is given here, but the detail in the experimental section is limited to flow rates and 'C18'. One assumes Teledyne materials were used, but details of column size, particle size and C18 sub-type must be provided in experimental (e.g. C18aq)

16) line 196: there is a typo here/word missing

17) line 200: pyrophosphate FORMING reaction

18) line 212: '...as THE major...'

19) Figure 4, clarify C=N in thiazolium structures.

20) The structure/atom label sizes in Figures 5 and 6 are rather small±please increase sizes.

21) line 231 '...cofactors ARE...'

22) lines 239-240: '...metabolized to A non-phosphorylated...'

23) Line 243: the mode of scrambling is not clear to me—is this referring to. Are the heavy atoms 2H or 18O, and is it a positional isotope exchange (i.e. bridging vs non-bridging O) or something else?

24) line 250: the last sentence is incomplete and very short—please rewrite/rephrase.

25) line 255: I would suggest replacing 'methylation', which implies the use of an "Me+"-type electrophilic reagent, with 'methyl estirifcation' or some similar phrase.

26) line 256: replace chromatography with chromatographic.

27) line 259: 100% D2O does not (as far as I am aware) exist. I imagine a 99.9% sample was used or similar—the specification+supplier should be given in the experimental detail.

28) line 265: this sentence doesn't really make sense—please recast.

29) line 275: hours-->h

30) line 276-277: triMethyl vs triEthyl.....there's some mix-up that needs fixing here.

31) line 285: '...the crude MATERIAL was...'

32) line 291: '...in detail here...'

33) line 303: '...challenges OF achieving...'

34) line 319: '...contentS of the jar WERE...'

35) line 322: 1H NMR data are mentioned here, whereas Table 1 mentions H and P NMR methods—which was used? Details should also be supplied in the SI and clarification between the use of H vs P should be given is both have been used across the examples.

36) lines 323-324 'trimethyl phosphate'

37) line 327: '...crude material was...'

38) lines 330-331: full details of the chromatography need to be supplied.

39) lines 373-382: this method is very intriguing, but it is not discussed at all. It should be given more detail, unless it is a known, literature protocol, in which case, appropriate literation citation(s) should be provided.

40) lines 386-395: Is this the method of APpy et al? If it is, please (re-)cite their work at this point.

41) line 415: Please give details of the chromatography protocol and/or cross reference to a general method provided at the beginning of the experimental section.

42) lines 505-506: pH has no meaning in acetone-MeOH mixture. A standard electrode cannot be used to give a numerical pH in this context. If an indicator paper was used, give details of the color that was produced—this does have a tangible, useful practical value of how  fare any neutralisation process has proceeded.

43) line 750: It's not clear what is meant by 'nuclear chemistry'—would 'isotopically labelled' or a similar set of words perhaps be more appropriate? 'Nuclear' implies radioisotopes, and if this were the meaning, this needs to be clarifies and explained.

44) Some of the spectra in the SI need attention because they do not deliver the information required to support the proposed structures:

pages 12-13: there is clear evidence for the presence of two components in these spectra. The experimental section does not report or advise on this impurity—this must be dealt with.

page 14+15: as above.

pages 17-19: fix C=N bonds in thiazolium ion structures.

pages 21-22: clear signs of heterogeneity that should be described in experimental.

page 24: fix C=N

page 31: This spectrum must be rerun—the 'water' signal fully obscures several of the sugar signals that need to be discerned. Also, discuss heterogeneity.

...and similar comments for some of the other spectra—report 'purity level' as estimated by H/P NMR.

page 84: The water signal is large and partially obscures sugar signals`–this really SHOULD be replaced with a better quality spectrum.

Author Response

We thanks the reviewer for their comments and suggestions. We have addressed the points individually as follows:

This is a very substantial piece of extremely useful synthetic work on very challenging systems. It was a pleasure to read through the work and I recommend 'minor corrections'. There are a few experimental details that must be tidied up, but, beyond these a some typos (listed below), this work very much deserves to be published and shared with the nucleotide community.

1) The title mentions only mechanochemistry and not the significant amount of solution-based work that is held within the manuscript.

Reply: We appreciate the reviewer’s comment and recognition of the value of the comparative study.  However, while solution phase chemistry, as used in the preparation of the phosphates, was compared to the mechanochemical routes, the synthesis of the dinucleotides, route which is usually the most challenging, was only pursued using mechanochemical approaches; hence the choice for the title.

2) page 1, line 17: "perfectly" is not a fair claim to make. 

Reply: corrected

3) page 1, line 33 cofactor, not cofactors…..

Reply: corrected

4) page 2, lines 82-85: the later portion of the manuscript and SI do not same to provide any substantiation of quantitative data to support degradation/chlorination outcomes, nor is literature evidence provided. Please provide supporting data.

Reply:  We have provided evidence in the ESI demonstrating that 4PYR was not phosphorylated under solution phase conditions and modified the text accordingly. The side-products that are generated upon forced conditions are not relevant to the present manuscript and therefore reference to their formation has been removed from the text.

5) The presentation of Figure 1 is a little unclear (which are R1, R2?) and a little messy (space lines at more regular angles, consistency of molecule spacings etc.

Reply: We have mended figure 1. However, with this figure we have attempted to provide the most encompassing way to describe the chemistry that we have performed without creating a complex figure.  B1 and B2 can be any of the reagents indicated in the figure.  We use the “red oxygen” in the reagents and the product as we wanted to describe how diverse the pool of product could be.  A bold oxygen might be required if the publication is black only.

6) In Figure 1 and throughout the manuscript the thiazolium structure is not very clear—the + charge seems to be displaced such that it obscures the C=N bond to the point that it is difficult to see—please rectify.

Reply: Corrected throughout the text.

7) page 3, line 131-133 and through the manuscript: use correct time abbreviations consistently i.e. h for hours (h is used in some places, hours elsewhere), min for minutes (mins is non-standard).

Reply: Corrected throughout the text.

8) page 4, line 135 '...unstuitable FOR riboflavin...'

Reply: Corrected

9) Figure 2: I found this (and most other figures) to be difficult to read because of the small text fonts (atom labels are ok). The table within Figure 2 is particularly hard to read. Please increase font size/consistency here and throughout the Figures, as needed.

Reply: All Figures have been revised throughout the text.

10) Figure 3—make C=N clear in all structures.

Reply: Corrected

11) Final entry of Table 1: The use of methanol is surprisingly described as NOT leading to methyl phsophoesters. Looking at the volumes of reagent/'solvent' used, this is a little surprising. Some degree of explanation and/or literature referencing, if this is a known process should be included.

Reply: We have described the synthesis of FMN in 4.1.3 Synthesis of FMN (32) and included reference [37]

12) line 168: each phosphoester (not plural)

Reply: Corrected

13) line 169: synthesEs (plural)

Reply: Corrected

14) line 173: 1 h, not 1 hr, and similar corrections for time units on lines 182, 185.

15) line 192: Some brief details of chromatography methodology is given here, but the detail in the experimental section is limited to flow rates and 'C18'. One assumes Teledyne materials were used, but details of column size, particle size and C18 sub-type must be provided in experimental (e.g. C18aq)

Reply: Details of chromatography methodology have been added in the experimental section (heading 4.3 General synthetic procedure for the preparation of pyrophosphates (20-26, 30a,31,33 &37) by mechanochemical ball-milling).

16) line 196: there is a typo here/word missing

Reply: Corrected

17) line 200: pyrophosphate FORMING reaction

Reply: Corrected

18) line 212: '...as THE major...'

Reply: Corrected

19) Figure 4, clarify C=N in thiazolium structures.

Reply: Corrected

20) The structure/atom label sizes in Figures 5 and 6 are rather small±please increase sizes.

Reply: Both the figures have been redrawn and resized.

21) line 231 '...cofactors ARE...'

Reply: Corrected.

22) lines 239-240: '...metabolized to A non-phosphorylated...'

Reply: Corrected

23) Line 243: the mode of scrambling is not clear to me—is this referring to. Are the heavy atoms 2H or 18O, and is it a positional isotope exchange (i.e. bridging vs non-bridging O) or something else?

Reply: Clarified

24) line 250: the last sentence is incomplete and very short—please rewrite/rephrase.

Reply:  Addressed in the text line 262.

25) line 255: I would suggest replacing 'methylation', which implies the use of an "Me+"-type electrophilic reagent, with 'methyl estirifcation' or some similar phrase.

Reply: Methylation has been replaced with methyl esterification. Now the line number is 263.

26) line 256: replace chromatography with chromatographic.

Reply: Corrected

27) line 259: 100% D2O does not (as far as I am aware) exist. I imagine a 99.9% sample was used or similar—the specification+supplier should be given in the experimental detail.

Reply: (a) we have corrected the percentage of D2O in the text. Now the  Line no is..267 (b) The details of D2O have been added in the section : (ii) Method B: Compound 41, line 572.

28) line 265: this sentence doesn't really make sense—please recast.

Reply: Corrected

29) line 275: hours-->h

Reply: Corrected throughout the text.

30) line 276-277: triMethyl vs triEthyl.....there's some mix-up that needs fixing here.

Reply: It was triethylamine not trimethylamine. Now it’s corrected

31) line 285: '...the crude MATERIAL was...'

Reply: Corrected

32) line 291: '...in detail here...'

Reply: Corrected

33) line 303: '...challenges OF achieving...'

Reply: Corrected

34) line 319: '...contentS of the jar WERE...'

Reply: Corrected

35) line 322: 1H NMR data are mentioned here, whereas Table 1 mentions H and P NMR methods—which was used? Details should also be supplied in the SI and clarification between the use of H vs P should be given is both have been used across the examples.

Reply: 1H-NMR method was used for table 1. The 31P-NMR method has been deleted from table 1. All the information related to table 1 has been updated in the supplementary files as well.

36) lines 323-324 'trimethyl phosphate'

Reply: Corrected

37) line 327: '...crude material was...'

Reply: Corrected

38) lines 330-331: full details of the chromatography need to be supplied.

Reply: Chromatographic detail has been in the section of 4.1.2 By solution-phase synthesis: Lines 339-347

39) lines 373-382: this method is very intriguing, but it is not discussed at all. It should be given more detail, unless it is a known, literature protocol, in which case, appropriate literation citation(s) should be provided.

Reply: 4..1.3 Synthesis of FMN (32): this is the patent method  (U. 8. Patent 2,610,178). The reference that describe the synthesis using this patent is now included in the references (ref 37)

40) lines 386-395: Is this the method of APpy et al? If it is, please (re-)cite their work at this point.

Reply: Reference number 36 has been added here in support of the preparation of monophospho-imidazolates (12-15) by mechanochemical ball-milling.

41) line 415: Please give details of the chromatography protocol and/or cross reference to a general method provided at the beginning of the experimental section.

Reply: The reference numbers 27 and 36 have been included in this section to aid with regards to information on the synthesis and purification.

42) lines 505-506: pH has no meaning in acetone-MeOH mixture. A standard electrode cannot be used to give a numerical pH in this context. If an indicator paper was used, give details of the color that was produced—this does have a tangible, useful practical value of how fare any neutralisation process has proceeded.

Reply: pH paper was used for the monitoring of the pH of the solution. The color of pH paper shifts from pink to yellow is added in brackets to support this comment.

43) line 750: It's not clear what is meant by 'nuclear chemistry'—would 'isotopically labelled' or a similar set of words perhaps be more appropriate? 'Nuclear' implies radioisotopes, and if this were the meaning, this needs to be clarifies and explained.

Reply: The statement has been rephrased for clarity

44) Some of the spectra in the SI need attention because they do not deliver the information required to support the proposed structures:

pages 12-13: there is clear evidence for the presence of two components in these spectra. The experimental section does not report or advise on this impurity—this must be dealt with.

Reply: A new sentence in support of query has been added in the text, section “Synthesis of C-5 deuterated adenosine monophosphate trimethylamine salt (49)”

page 14+15: as above.

Reply: A new sentence in support of query has been added in the text, , section “Synthesis of [18O] NMN (15):”

pages 17-19: fix C=N bonds in thiazolium ion structures.

Reply: Corrected throughout the text and supplementary file.

pages 21-22: clear signs of heterogeneity that should be described in experimental.

Reply: Heterogeneity has been explained in supplementary file (SI). Now the current page number is 17 and 19 in SI.

page 24: fix C=N

Reply: Done

page 31: This spectrum must be rerun—the 'water' signal fully obscures several of the sugar signals that need to be discerned. Also, discuss heterogeneity.

...and similar comments for some of the other spectra—report 'purity level' as estimated by H/P NMR.

Reply: Revised accordingly.

page 84: The water signal is large and partially obscures sugar signals`–this really SHOULD be replaced with a better quality spectrum.

Reply: revised accordingly.

Round 2

Reviewer 1 Report

Most of my concerns have been addressed. The revised manuscript can be considered for publication after minor revisions.

  • “24h” should be “24 h” (a space is required) in Table 1. Other similar problems in Table 1 and other places should be corrected throughout the manuscript.
  • The reaction time can be all expressed in hours. e.g. 90 min can be changed to 1.5 h in Table 2.

Reviewer 2 Report

The revised ms is acceptable.